# Whole-Genome Resequencing Reveals the Genetic Diversity and Selection Signatures of the *Brassica juncea* from the Yunnan-Guizhou Plateau

Xiaoyan Yuan [1,2], Minglian Fu [2], Genze Li [2], Cunmin Qu [3], Hao Liu [1], Xuan Li [1], Yunyun Zhang [2], Yusong Zhang [2], Kaiqin Zhao [2], Lifan Zhang [2], Yanqing Luo [2], Jinfeng Li [2], Xiaoying He [2], Liu He [1] and Feihu Liu [1,*]

1 School of Agriculture, Yunnan University, Kunming 650091, China
2 Industrial Crop Institute, Yunnan Academy of Agricultural Sciences, Kunming 650200, China
3 College of Agronomy and Biotechnology, Southwest University, Chongqing 400715, China
* Correspondence: liufh@ynu.edu.cn

**Abstract:** *Brassica juncea* has adapted to diverse climate zones and latitudes, especially in the Yunnan-Guizhou Plateau with the complexity and diversity of the ecological types of western China. However, the genetic variations underlying the diversity of these ecotypes are poorly known. In this study, we resequenced the genome of 193 indigenous *B. juncea* accessions and obtained 1.04 million high-quality SNPs and 3.23 million InDels by mapping reads to the reference genomes of *B. juncea* var. *timuda*. Phenotype, population genetic, phylogenetic and principal component analyses showed considerable genetic variation including four distinct genetic groups. Selective sweep analysis and a genome-wide association study revealed the candidate genes for seed color and fatty acid biosynthesis. The results provide a comprehensive insight into the spread and improvement of *B. juncea* and laya foundation for accelerating rapeseed breeding by facilitating screenings of molecular markers.

**Keywords:** *Brassica juncea*; GWAS; seed coat color; fatty acid

## 1. Introduction

The mustard *Brassica juncea* (L.) (Czern and Coss, AABB, 2n = 36) originated from an interspecific hybridization of the diploid species *Brassica rapa* (AA, 2n = 20) and *Brassica nigra* (BB, 2n = 16) [1,2]. It is also regarded as one of the earliest domesticated plants, firstly used as a condiment in Sanskrit and Sumerian texts from as early as 3000 BC [3]. The records of its center origin are diverse. For example, Central Asia usually acted as the primary origin center of *B. juncea*, and central/western China and eastern India were secondary centers of diversity [4], while many studies showed that *B. juncea* should have evolved in the Middle East [5,6]. Recently, some researchers supported that China was the primary origin and diversity center of *B. juncea* [2,7]. In any case, as one of the important origin centers of *B. juncea*, China is rich in *B. juncea* germplasm, especially in the Yunnan-Guizhou Plateau.

To date, *B. juncea* has been widely grown as oilseed, condiment, and vegetable crops for human consumption [8]. As we know, canola oil, the third largest edible oil in the world, comes from one of three species: *B. napus*, *B. rapa*, or *B. juncea* [9]. Among them, *B. napus* is the second most important oilseed crop in the world, whereas *B. juncea* also has many valuable agronomic characteristics, which are exactly what other *Brassica* species need, including more vigorous seedling growth, early maturity, tolerance to abiotic stresses, and yellow seeds [10,11]. Moreover, the yellow trait and blackleg resistance had been used for improving *B. napus* by interspecific crossing [12,13]. In addition, the quality characteristics of *B. juncea*, including the fatty acid profile and low glucosinolate content in the meal, were further improved through an interspecific cross with *B. napus* and *B. rapa* [14,15]. In *Brassica* species, the yellow-seeded varieties are mostly preferred over brown-seeded ones

due to their high percentage of oil and protein, lower content of fiber, and high-quality feed for livestock. *Brassica napus* is the largest oil crop in China. Unfortunately, no natural yellow-seeded germplasms have been found in *B. nupus*, but many natural yellow-seeded germplasms were found in *B. juncea*. Through long-term interspecific hybridization and systematic breeding, yellow-seeded genes were introduced from *B. juncea* to *B. napus*, and many yellow-seeded *B. napus* varieties were commercially cultivated [11,16].

However, there are still disputes about the pigment substances that affect the seed coat color of rapeseed. Marles et al. identified proanthocyanidins as the main pigments affecting the seed coat color [17]. Xiang et al. found that procyanidins, quercetin, and lignin have a great impact on the color of the seed coat, while the brown seed coat does not contain kaempferol [18]. Moreover, the information about the candidate gene(s) involved in the yellow trait of oilseed *Brassica species* is very limited. To date, only genes involved in the flavonoid biosynthesis pathway that play important roles in seed coat color formation have been cloned, such as *TTG1*, *TT8*, and *TT1* [19–23]. The molecular genetic mechanism of the yellow trait is almost unknown in *Brassica* species. If we obtain the gene that controls the yellow seed coat characteristic of *B. juncea*, it will play an important role in increasing the oil yield and improving the quality.

Fatty acid composition is one of the most important characteristics of oil crops because it determines the quality of edible oil. The main fatty acid composition of rapeseed oil includes saturated fatty acid stearic acid (C18: 0), monounsaturated fatty acid oleic acid (C18: 1), polyunsaturated fatty acid linoleic acid (C18: 2) and linolenic acid (C18: 3), ultra-long chain saturated fatty acid eicosenoic acid (C20: 1), and erucic acid (C22: 1). Erucic acid is not suitable for food processing because its long carbon chain is not easily digested and absorbed in the human body [24]. Therefore, one of the main objectives of edible rapeseed breeding is to reduce erucic acid content (EAC). In the 1950s and 1960s, Canada took the lead in creating the world's first low-erucic-acid rapeseed variety, Oro [25]. Since then, low-erucic-acid rapeseed varieties have emerged in many countries. However, the allelic resources of low-erucic-acid rapeseed currently used are only from a few parents such as Oro, which leads to a relative lack of breeding resources. In addition, erucic acid is an important industrial material, which can be used as high-grade lubricating oil for steel casting, aerospace, navigation, and other industries. In nature, erucic acid only exists in a few plant seeds with relatively high content. At present, plant erucic acid in the world mainly comes from high-erucic-acid rapeseed and sea cabbage [26]. Increasing the EAC of rapeseed can significantly reduce the production cost and increase its market prospects. Most studies believe that fatty acid elongase-1 (*FAE1*) is a key rate-limiting enzyme for erucic acid synthesis [27–29], which has been confirmed in many crops [30–35]. β-Ketoacyl-CoA synthase (*KCS*) is encoded by the fatty acid elongase-1 (*FAE1*) gene.

In *B. juncea*, there are many excellent characteristics that contribute a valuable genetic resource for improving *Brassica* species. The Yunnan-Guizhou Plateau is one of the districts kept the highest genetic diversity of cultivated *B. juncea* in China [36]. In this study, we sequenced 193 *B. juncea* accessions to examine their intraspecific genetic diversification and detected a series of candidate genes for seed coat color and erucic acid by using selective sweeps and genome-wide association studies (GWASs). The results built a database covering the genetic diversity of *B. juncea* accessions from the Yunnan-Guizhou Plateau and laid a foundation that underpins the genetic improvement of *Brassica* oil and vegetable crops.

## 2. Materials and Methods

### 2.1. Plant Materials and Phenotyping

A total of 193 *B. juncea* accessions were collected from the major breeding institutes at the Yunnan Academy of Agricultural Sciences, Kunming, Yunnan, China, including 186 from China, 6 from India, and 1 from Poland. All *B. juncea* accessions were maintained by self-pollination for at least five generations at Industrial Crop Institute, Yunnan Academy of Agricultural Sciences, Kunming, Yunnan, China. The detailed information of these accessions, including variety name and country of origin, is listed in Table S1. All accessions

were planted in field trials in Kunming, Yunnan, China (102.10° E, 26.22° N) in 2019, 2020, and 2021 growing seasons with three replications. Seeds were sown at the beginning of October and harvested in the following May. Each accession was grown in two rows with 10 plants. Mature seeds were randomly collected from at least 5 plants for seed coat color and fatty acid analysis, and fully mature and plump seeds were used for discerning the seed coat color of each accession. Oleic acid, linoleic, linolenic acid, erucic acid, palmitic acid, stearic acid, and arachidonic acid were detected by near-infrared reflectance spectroscopy (FOSS NIPSystem 461040 (FOSS, Minnesota, USA)). Descriptive statistics, analysis of variance (ANOVA), and correlation analysis of phenotypic data were performed using IBM SPSS v24 (https://www.ibm.com/cn-zh/spss (accessed on 3 September 2022)).

### 2.2. Resequencing and Discovery of Genomic Variations

Genomic deoxyribonucleic acid (DNA) was isolated from young leaves of each accession using the DNAquick Plant System Kit (TIANGEN, Beijing, China) following the standard protocol. The fragments with a mean insert size of 350 bp were gel-excised and eluted. The DNA libraries were amplified and 150-bp paired-end reads were generated on an Illumina NovaSeq6000 instrument (Illumina, CA, USA). Library preparation and sequencing were carried out at the Lc-Bio Technologies limited company (Hangzhou, China).

A total 1.45Tb of clean data was generated after removing reads with ≥5% unidentified nucleotides (N), >10 nucleotides aligned to the adaptor, or of which >10% bases had Phred quality scores less than 30. The paired-end reads were mapped to the *B. juncea* var. *timuda* version 1.5 reference genome [2] using BWA v0.7.13 [37]. The genomic variants for each accession were then identified by Genome Analysis Toolkit software (https://gatk.broadinstitute.org/ (accessed on 8 October 2022)) and SAMtools v0.1.19 [38]. All the GVCF files were merged. The high-quality single-nucleotide polymorphisms (SNPs) and insertion–deletions (InDels) were created with the following parameters: depth for individual ≥6.5, minor allele frequency (MAF) ≥0.05, missing rate ≤0.2.

### 2.3. Population Structure and Phylogenetic Analyses

The population genetic structure was examined using the program Admixture v1.30 [39] with K values from 1 to 10. K = 7 was chosen because the clusters maximized the marginal likelihood. Phylogenetic tree analysis was performed using MEGA v1.6.6 (https://www.megasoftware.net/ (accessed on 26 October 2022)) using the neighbor-joining algorithm (NJ) model. The reliability of the NJ trees was estimated using the bootstrapping method with 1000 replicates. The genetic relationship and principal component analysis (PCA) were performed using the smartPCA program from the EIGENSOFT package v.6.0.1 (https://github.com/DReichLab/EIG (accessed on 6 December 2022)).

To estimate and compare the patterns of linkage disequilibrium (LD) among different genetically distinct groups, the correlation coefficients ($r^2$) between pairwise SNPs were computed using plink2 software (https://www.cog-genomics.org/plink/2.0/ (accessed on 12 October 2022)). LD decay statistics were calculated for different subgroups, and LD decay graphs were plotted using PopLDdecay v3.26 (https://github.com/BGI-shenzhen/PopLDdecay (accessed on 15 December 2022)), with pairwise markers in a 50-kb window and averaged across the whole genome. The kinship matrix (K) of the association population was calculated using TASSEL 5.2.1 (https://www.maizegenetics.net/tassel (accessed on 29 December 2022)).

### 2.4. Genome-Wide Selective Sweep Analysis and Selection of Candidate Genes

In order to detect candidate regions potentially affected by selections, the nucleotide diversity ($\pi$) and fixation index ($F_{ST}$) were calculated using vcftools v0.1.13 (https://vcftools.github.io (accessed on 26 December 2022)) [40], and pairwise genetic distance was calculated by Arlequin v3.5.2.2 (http://cmpg.unibe.ch/software/arlequin3/ (accessed on 26 December 2022)) in a 100-kb sliding window with a step size of 10 kb. The value of $\pi$ reflects the nucleotide diversity of the population genome. $F_{ST}$ indicates the degree of

segregation between genetically distinct groups, which is the inbreeding coefficient of a subpopulation. According to the $F_{ST}$ value, the degree of genetic differentiation between populations is divided into four categories: small, medium, large, and extra-large, and the corresponding $F_{ST}$ values of the four categories are 0–0.05, 0.05–0.15, 0.15–0.25, and more than 0.25, respectively [41]. The $\theta_\pi$ ratio values were calculated based on the ratio of $\theta_\pi$ for a genetically distinct group with respect to a control group. Selective sweep regions between different genetically distinct groups were selected according to the interception of the two parameters $F_{ST}$ and $\theta_\pi$ ratio. The regions with the top 5% $F_{ST}$ and $\theta_\pi$ ratio values simultaneously were considered as candidate outliers under strong selective sweeps. The annotated genes residing in these regions were considered as candidate genes.

*2.5. GWAS and Identification of Candidate Genes*

In this study, only SNPs with MAF $\geq$ 0.05 and missing rate $\leq$0.1 in a population were used in our GWAS. A total of 3,972,292 high-quality SNPs were used for the GWAS. The top three PCs were used to construct the Q matrix for population-structure correction. The K-value, which represents the genetic relations between samples, was calculated by SPAGeDi software (https://ebe.ulb.ac.be/ebe/SPAGeDi.html (accessed on 20 December 2022)). The GWAS were performed on the SNP set using mixed linear modeling implemented in the EMMAX program (Efficient Mixed-Model Association eXpedited, https://csg.sph.umich.edu/kang/emmax/ (accessed on 5 January 2023)). The significant thresholds of all tested traits were evaluated with the formula P = 0.05/n (where n is the number of SNPs). Then, 100kb sequence regions adjacent to the significantly associated SNPs were searched for associated genes. All candidate genes were mapped to GO terms in the Gene Ontology database (http://www.geneontology.org/ (accessed on 30 December 2022)). Significantly enriched GO terms were defined using a hyper geometric test with *p* < 0.05. KEGG is a major public pathway-related database. The calculation formula is the same as that in GO analysis. Pathways with *p* < 0.05 were defined as significantly enriched pathways in genes.

## 3. Results

### 3.1. Genotype of B. juncea

To explore genetic variation in *B. juncea*, we resequenced the 193 accessions and obtained 1.45 Tb of clean data with an average depth of 10×. The mapping rate varied from 92.32 to 94.27%. Using a stringent SNP-calling protocol, a total of 8,851,861 high-quality SNPs and 3,233,088 InDels were identified (Table S2). The numbers of SNPs and InDels were unevenly distributed on chromosomes of *B. juncea*, and the lowest and the highest chromosomes were B08 and A10, respectively (Figure S1). A total of 20.9% of SNPs and 5.27% of InDels were located in the coding region, while 13.1% of SNPs caused the missense variant, 7.74% of SNPs resulted in the loss or gain of stop codons, and 13.44% of InDels led to frameshift mutations.

### 3.2. Population Structure of B. juncea

The population structure of 193 accessions was investigated using the software Admixture. When the K values were from 4–7, the cross-validation error values were relatively low and the differences between them were small (Figure 1a). Most samples of genetically distinct groups should have extensive gene exchanges with other groups (Figure 1b). To better clarify the relationships within the population, 47 genetically admixed accessions with main genetic components of less than 60% were excluded from further analysis. Both phylogenetic and principal component analyses (PCAs) of the remaining accessions indicated four distinct groups. Furthermore, 146 accessions were divided into four genetically distinct groups (G1–G4), including 30 accessions in G1, 25 in G2, 55 in G3, and 36 in G4, respectively, based on population structure, principal component analyses, phylogenetic analysis, and phenotype analysis (Figure 1b).

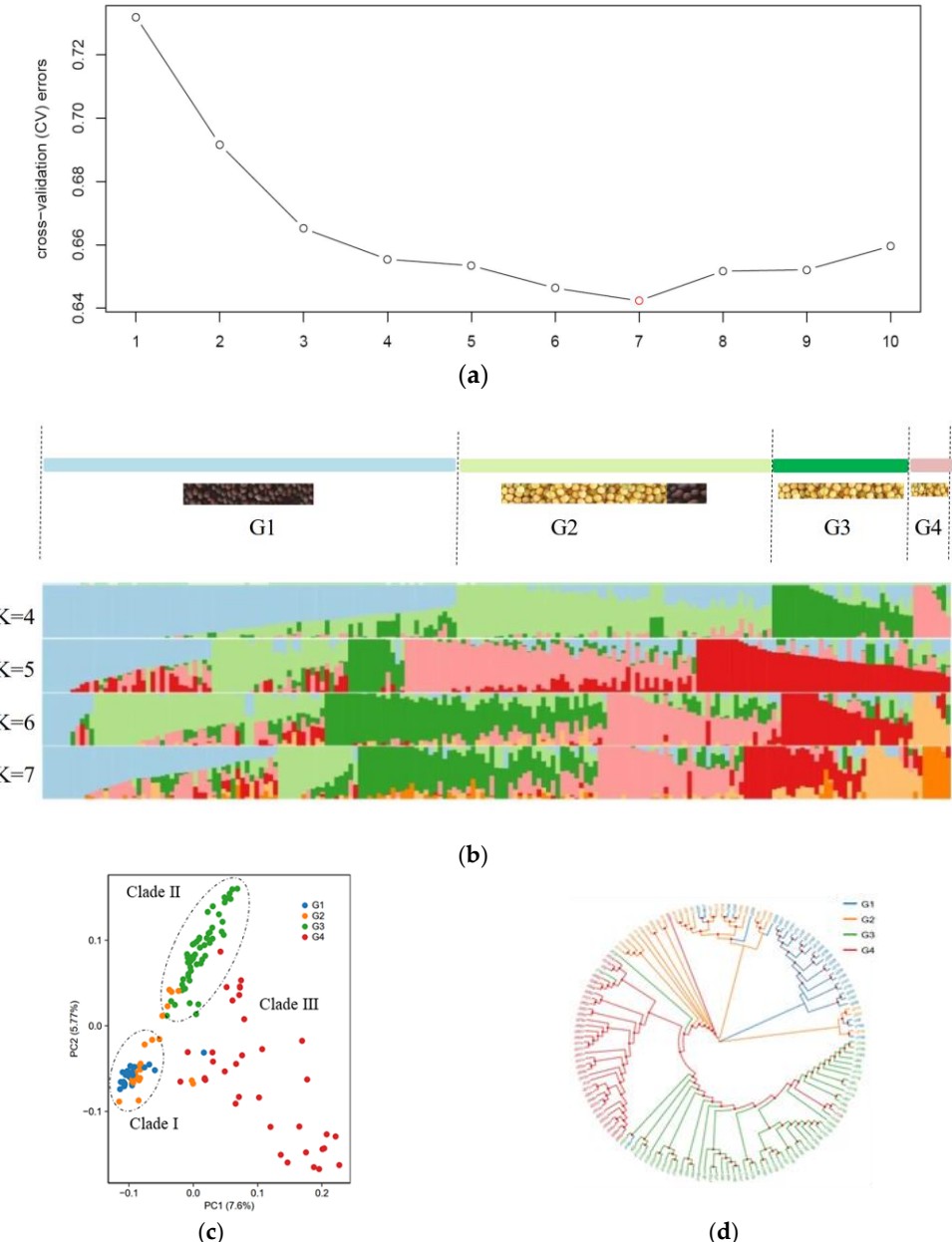

**Figure 1.** Genetic diversity and population structure of the studied *B. juncea* accessions. (**a**) ADMIX-TURE was used to analyze the population structure of accessions, and the cross-validation error was calculated when K was 1–10. (**b**) The population structure when K was 2–7. Each individual is represented by a line with different colors, and the group to which the individual belongs is inferred based on the proportion of the color. The seven color regions arranged from left to right in order to represent G1 to G4, respectively. (**c**) PCA of the group; one point represents an individual. Clades with bootstrap values of above 50% are indicated by a circled red dot. (**d**) Phylogenetic tree of the group.

Results from the PCA showed that 146 *B. juncea* accessions have relatively scattered spatial distribution characteristics, and the contribution rates of principal components PC1 and PC2 were 7.6% and 5.77%, respectively, indicating that their genetic background was relatively extensive. The PCA shows that G2 is closest to G1 (Figure 1c), which is consistent with the result of constructing the phylogenetic tree based on the adjacency method (Figures 1d and S2). Most of the accessions from G1 and G2 were grouped in clade I and accessions from G3 in clade II. The rest of the accessions were clustered in clade

III (Figure 1c). This was largely in accordance with those of the morphological type of division. For example, G1 contained mostly the brown-seeded accessions, while G2, G3, and G4 contained only the yellow-seeded accessions (Table S1, Figure S5). In addition, the content of long-chain saturated fatty acids in G4, including erucic acid content (EAC) and arachidonic acid content, were significantly lower than that in G1–G3, while the content of unsaturated fatty acids, such as oleic acid, linoleic acid, and linolenic acid, were opposite (Table 1).

**Table 1.** Mean value of some agronomic characters in each genetically distinct group.

| Trait | Group | Mean | SD | Max. | Min. | CV (%) | Skewness | Kurtosis | G | E | h² (%) |
|---|---|---|---|---|---|---|---|---|---|---|---|
| Stearic acid (%) | G1 | 3.779 | 0.412 | 5.235 | 3.178 | 10.900 | 4.139 | 1.560 | ** | ** | 93.390 |
| | G2 | 3.687 | 0.576 | 4.714 | 2.669 | 15.621 | −0.661 | 0.212 | | | |
| | G3 | 3.845 | 0.456 | 4.799 | 2.705 | 11.866 | 0.152 | −0.022 | | | |
| | G4 | 5.187 | 1.168 | 11.531 | 4.324 | 22.517 | 26.341 | 4.818 | | | |
| Palmitic acid (%) | G1 | 0.995 | 0.152 | 1.339 | 0.714 | 15.302 | 0.359 | 0.315 | ** | ** | 87.640 |
| | G2 | 1.102 | 0.151 | 1.464 | 0.763 | 13.671 | 1.141 | 0.380 | | | |
| | G3 | 1.126 | 0.119 | 1.474 | 0.710 | 10.537 | 2.717 | −0.499 | | | |
| | G4 | 1.375 | 0.238 | 2.455 | 1.082 | 17.284 | 11.608 | 2.734 | | | |
| Oleic acid (%) | G1 | 19.773 | 5.619 | 34.944 | 9.295 | 28.419 | 0.440 | 0.339 | ** | ** | 87.070 |
| | G2 | 17.328 | 6.332 | 30.210 | 2.071 | 36.541 | 0.983 | −0.085 | | | |
| | G3 | 19.776 | 5.942 | 38.429 | 9.714 | 30.046 | 0.736 | 0.822 | | | |
| | G4 | 52.561 | 4.766 | 72.017 | 44.181 | 9.067 | 7.206 | 1.811 | | | |
| Linoleic acid (%) | G1 | 19.768 | 1.338 | 23.051 | 17.133 | 6.770 | 0.440 | 0.273 | ** | ** | 87.830 |
| | G2 | 19.834 | 1.510 | 22.432 | 16.888 | 7.614 | −0.266 | −0.174 | | | |
| | G3 | 19.979 | 1.590 | 25.559 | 16.081 | 7.961 | 2.079 | 0.425 | | | |
| | G4 | 29.966 | 1.811 | 33.458 | 26.466 | 6.042 | −0.893 | 0.024 | | | |
| Linolenic acid (%) | G1 | 12.480 | 1.055 | 14.572 | 10.292 | 8.455 | −0.099 | −0.365 | ** | | 86.820 |
| | G2 | 12.754 | 0.893 | 14.464 | 11.212 | 7.000 | −0.848 | 0.189 | | | |
| | G3 | 12.504 | 0.885 | 14.722 | 10.215 | 7.079 | 0.424 | −0.186 | | | |
| | G4 | 16.885 | 1.940 | 25.399 | 13.211 | 11.489 | 10.346 | 2.099 | | | |
| Arachidonic acid (%) | G1 | 8.820 | 1.912 | 12.296 | 4.508 | 21.680 | −0.596 | −0.218 | ** | ** | 94.960 |
| | G2 | 9.017 | 2.039 | 12.826 | 3.450 | 22.619 | 1.543 | −0.676 | | | |
| | G3 | 8.639 | 2.011 | 12.005 | 0.001 | 23.274 | 5.050 | −1.571 | | | |
| | G4 | 0.474 | 0.732 | 3.559 | 0.001 | 54.342 | 8.026 | 2.471 | | | |
| Erucic acid (%) | G1 | 31.064 | 4.590 | 39.020 | 18.781 | 14.775 | 1.297 | −0.857 | ** | ** | 87.380 |
| | G2 | 30.633 | 5.565 | 42.222 | 17.130 | 18.168 | 1.088 | −0.391 | | | |
| | G3 | 29.296 | 4.940 | 38.920 | 11.492 | 16.862 | 1.885 | −0.873 | | | |
| | G4 | 0.002 | 0.001 | 0.004 | 0.001 | 36.464 | −0.337 | 0.791 | | | |

Note: ** The values are significant at $p < 0.01$ for the effect of genotype (G) and environment (E) on phenotypic variance estimated by one-way ANOVA. SD, standard deviation; CV, coefficient of variation; h², broad-sense heritability.

### 3.3. Genetic Diversity of B. juncea

The nucleotide diversity index (π) was calculated by SNPs to evaluate the genetic diversity of the accessions. The π value of G1 was the highest, G2 and G3 were low, and G4 was the lowest, indicating that G1 maybe relatively more primitive. Other groups may diverged from it (Table S3). The selection of different agronomic traits occurred with the diversion of genetically distinct groups, such as the color of the seed coats from brown to yellow; the long-chain saturated fatty acids represented by erucic acid from high to low; and the monounsaturated fatty acids, such as oleic acid and polyunsaturated fatty acids, linoleic acid, and linolenic acid from low to high (Table 1).

We compared the linkage disequilibrium (LD) decay of different groups (Figure S3). At a threshold of $r^2 = 0.2$, the LD decays in G1 (23.96 kb), G2 (34.74 kb), and G3 (32.94 kb) were stronger than those in G4 (69.83 kb), supporting a strong human selection during *B. juncea* spread and improvement and G1 being relatively primitive in four genetically distinct groups.

### 3.4. Kinship of B. juncea

The kinship value between two pairs of 193 accessions were calculated. The kinship value equal to 0 accounted for 60.52%, less than 0.5 accounted for 94.47%, and more than 0.5 accounted for only 0.13%. The kinship between two pairs of 193 accessions were generally far (Figure S4), and only a few of them were relatively close.

In order to accurately valuate the genetic differentiations and relationships among different groups, we also calculated the fixation index ($F_{ST}$) between pairs of groups (Table S3). G1 and G2 had the minimum genetic differentiation and the closest genetic relationship ($F_{ST}$ = 0.029; Figure 1c). Secondly, G3 was moderately differentiated from G1 and G2 (pairwise $F_{ST}$ = 0.094 and 0.081, respectively). G4 was largely differentiated from G1 (pairwise $F_{ST}$ = 0.120) and moderately differentiated from G2 and G3 (pairwise $F_{ST}$ = 0.109 and 0.084, respectively; Table S3), showing that the genetic differentiation between clade I and clade III was significant.

Based on the aforementioned results and the morphological differences of different groups, we could speculate that G1 was the original *B. juncea* in the Yunnan-Guizhou Plateau and that its improvement process had probably gone through two stages. The first stage (stage 1) was the process from brown-seeded accessions to yellow-seeded accessions (from clade I to clade II), and the second stage (stage 2) was the process from high long-chain saturated fatty acids to high unsaturated fatty acids (from clade II to clade III). These results are consistent with historical breeding records in the last century that involved extensively breeding low-erucic-acid and yellow-seeded varieties to replace high-erucic-acid and brown-seeded varieties.

Nucleotide diversity ($\pi$) decreased from $1.58 \times 10^{-3}$ to $1.47 \times 10^{-3}$ in stage 1 and from $1.47 \times 10^{-3}$ to $1.16 \times 10^{-3}$ in stage 2 (Table S3), implying that during stage 2, more genetic diversity was lost. In addition, G3 differentiated less strongly from G2 (pairwise $F_{ST}$ = 0.081) than G4 from G3 (pairwise $F_{ST}$ = 0.084; Table S3), suggesting that more genetic differentiation occurred in stage 2 than stage 1. This large genetic differentiation may have mainly resulted from the most important event in the history of rapeseed breeding in the last century, that is, the widespread use of low-erucic-acid varieties Altex to improve the original high-erucic-acid varieties.

### 3.5. Candidate Genes for Seed Coat Color

We observed seed coat color variation across the 193 accessions grown in three years (Figure S5). The seed coat colors of the 193 accessions were brown or yellow and were consistent in the three years. Brown-seeded accessions were only distributed in G1, while yellow-seeded materials were mainly distributed in G2, G3, and G4 (Table S1).

The main characters selected in stage1 were the seed coat colors from brown to yellow. Based on the comprehensive evaluation of $F_{ST}$ and the ratio of nucleotide diversity ($\theta_\pi$ ratio), the 5% intersection of the two indexes was taken to select the signal, resulting in $F_{ST}$ (G1/G2) > 0.13, $\theta_\pi$ ratio (G4/G1) $\geq$ 1.82, $F_{ST}$ (G1/G3) > 0.31, and $\theta_\pi$ ratio (G1/G3) $\geq$ 2.03. We identified 115 and 96 putative selective sweeps in G1/G2 andG1/G3, respectively, containing 2089 and 1923 candidate genes (Table S4). A total of 542 genes were conserved between more than two distinct group differentiation regions, 36.90% (200) of which were distributed on A09, 43.17% (234) on B08. Of these genes, fifteen have known roles in the flavonoid biosynthesis pathway (Table 2) and two genes (BjuB018407 and BjuA034936) also showed significant association with being yellow-seeded by GWAS analysis (Table S5).

**Table 2.** Genes related to the flavonoid synthesis pathway.

| Gene ID | Chr. | Length | Start | End | Annotation | Selection Groups |
|---------|------|--------|-------|-----|------------|------------------|
| BjuB018407 | B08 | 915 | 51,641,320 | 51,642,396 | Flavonol synthase/flavanone 3-hydroxylase | G1/G2, G1/G3 |
| BjuA034859 | A09 | 846 | 32,483,465 | 32,484,597 | Heme oxygenase 1 | G1/G2, G1/G3 |
| BjuA034858 | A09 | 351 | 32,474,088 | 32,474,438 | Biliverdin-producing, ferredoxin | G1/G2, G1/G3 |
| BjuB019589 | B03 | 1332 | 43,953,714 | 43,955,125 | Cytokinin-N-glucosyltransferase | G1/G2, G1/G3 |
| BjuA040986 | A02 | 1083 | 1,316,124 | 1,318,173 | Flavonol synthase | G1/G2 |
| BjuA025255 | A07 | 2475 | 1,472,948 | 1,477,830 | C-type lectin domain family 16, member A | G1/G2 |
| BjuA036020 | A07 | 1290 | 806,774 | 808,218 | Shikimate O-hydroxycinnamoyltransferase | G1/G2 |
| BjuA034936 | A09 | 861 | 27,359,538 | 27,361,184 | TRANSPARENT TESTA 1 | G1/G2 |
| BjuA035531 | A09 | 1896 | 42,739,146 | 42,742,220 | Chalcone isomerase | G1/G2 |
| BjuA036650 | A09 | 1491 | 49,057,032 | 49,059,418 | Isoflavone/4′-methoxyisoflavone 2′-hydroxylase | G1/G2 |
| BjuB003779 | B02 | 1008 | 48,552,117 | 48,553,791 | Flavonol synthase/flavanone 3-hydroxylase | G1/G2 |
| BjuB004510 | B02 | 1011 | 48,559,974 | 48,561,651 | Flavonol synthase/flavanone 3-hydroxylase | G1/G2 |
| BjuA034990 | A09 | 633 | 34,170,686 | 34,172,497 | Shikimate O-hydroxycinnamoyltransferase | G1/G3 |
| BjuA036409 | A09 | 756 | 47,855,094 | 47,856,199 | Fatty-acid-binding protein 1 | G1/G3 |
| BjuB029287 | B04 | 1077 | 19,998,112 | 20,000,046 | Isoflavone/4′-methoxyisoflavone 2′-hydroxylase | G1/G3 |

We detected two missense variants in the first intron of BjuB018407 flavonol synthase (*FLS*) that were significantly associated with the seed coat color. We found that most yellow-seeded accessions (129/134) had these two missenses, whereas they were carried only by some brown-seeded accessions (25/59). The first missense variant site was 5 bp away from the intron splicing site. *FLS* gene is one of the most important node genes at the junction of flavonoid synthesis, which not only affects the synthesis of flavonoids, but also the accumulation of anthocyanin and finally determines the color of the flower or seed coat [42]. Flavonol and anthocyanin glycosides share the previous synthesis pathway, and FLS in flavonoid synthesis pathway and dihydroflavonol 4 reductase (DFR) in the anthocyanin synthesis pathway competes for the same substrate [43].

### 3.6. Major Genes for Fatty Acid Composition

We determined the fatty acid composition of 193 accessions grown in three years (Table S6 and Figure S6). Significant positive correlations were detected in the results of the three years, with R values of 0.74–0.99 (Table S6). During stage 2, the content of saturated long-chain fatty acids declined, especially erucic acid, while the content of unsaturated fatty acids including oleic acid, linoleic acid, and linolenic acid increased (Table 1).

Based on the comprehensive evaluation of $F_{ST}$ and the $\theta_\pi$ ratio, the 5% intersection of the two indexes was taken to select the signal, resulting in $F_{ST}$ (G3/G4) > 0.29, $\theta_\pi$ ratio (G3/G4) ≥ 1.98, $F_{ST}$ (G2/G4) > 0.36, $\theta_\pi$ ratio (G2/G4) ≥1.95, and $F_{ST}$ (G3/G7) > 0.56. We found 81 putative selective sweeps in G3/G4 andG2/G4 respectively, containing 1839 and 2175 candidate genes (Table S7). A total of 1204 genes were conservative, of which 85.80% (1033) were distributed on A08. Eighteen genes are known to be related to erucic acid metabolism (Table 3), of which the first ten genes were concurrently selected in the three repetitions across three years by GWAS analysis (Figure S7 and Table S8).

**Table 3.** Genes associated with fatty acid metabolism.

| Gene ID | Chr. | Length | Start | End | Annotation | Selection Groups |
|---------|------|--------|-------|-----|------------|------------------|
| BjuA046600 | A08 | 876 | 10,993,813 | 10,995,009 | SNF1-related protein kinase regulatory subunit beta-2 | G2/G4, G3/G4 |
| BjuA029874 | A08 | 1215 | 20,449,210 | 20,451,344 | Pyruvate dehydrogenase | G2/G4, G3/G4 |
| BjuA029561 | A08 | 885 | 18,709,038 | 18,709,922 | Fatty acid desaturase 4 | G2/G4, G3/G4 |
| BjuA029559 | A08 | 885 | 18,701,624 | 18,702,508 | Fatty acid desaturase 4 | G2/G4, G3/G4 |
| BjuA029009 | A08 | 1341 | 17,215,938 | 17,218,074 | Omega-6 fatty acid desaturase | G2/G4, G3/G4 |
| BjuA028960 | A08 | 1677 | 16,923,302 | 16,927,478 | 3-hydroxyisobutyryl-CoA hydrolase-like protein | G2/G4, G3/G4 |
| BjuA028763 | A08 | 774 | 14,533,575 | 14,534,960 | Short-chain dehydrogenase/reductase SDRA | G2/G4, G3/G4 |
| BjuA028614 | A08 | 798 | 13,046,485 | 13,048,555 | Palmitoyl-protein thioesterase 1 | G2/G4, G3/G4 |
| BjuA028208 | A08 | 903 | 12,546,658 | 12,548,359 | Probable enoyl-CoA hydratase 2 | G2/G4, G3/G4 |
| BjuA028013 | A08 | 1065 | 2,947,736 | 2,949,499 | Electron transfer flavoprotein subunit alpha | G2/G4, G3/G4 |
| BjuA000690 | A08 | 798 | 13,120,281 | 13,122,367 | Palmitoyl-protein thioesterase 1 | G2/G4, G3/G4 |
| BjuA000215 | A08 | 1338 | 17,360,042 | 17,362,175 | Omega-6 fatty acid desaturase | G2/G4, G3/G4 |
| BjuB047302 | B03 | 1518 | 8,750,374 | 8,751,891 | 3-ketoacyl-CoA synthase 18 | G2/G4, G3/G4 |
| BjuB047301 | B03 | 1422 | 8,747,481 | 8,748,902 | 3-ketoacyl-CoA synthase 17 | G2/G4, G3/G4 |
| BjuB032119 | B03 | 1425 | 8,975,890 | 8,977,314 | 3-ketoacyl-CoA synthase 17 | G2/G4, G3/G4 |
| BjuB032118 | B03 | 1518 | 8,978,894 | 8,980,411 | 3-ketoacyl-CoA synthase 18 | G2/G4, G3/G4 |
| BjuB032028 | B03 | 1008 | 9,742,346 | 9,743,766 | Short-chain dehydrogenase/reductase | G2/G4, G3/G4 |
| BjuB032027 | B03 | 771 | 9,744,629 | 9,745,972 | Short-chain dehydrogenase/reductase | G2/G4, G3/G4 |

Notably, a one bp insertion in the exon of BjuB032118 (*FAE1*) was found to be significantly associated with the content of erucic acid (Figure 2a,b). This insertion may have resulted in a code shift variant and obtained a stop code. *FAE1* gene is known as a key gene coded Ketoyl CoA Synthase [44] that regulates the synthesis of ultra-long chain fatty acids such as erucic acid [45]. Two haplotypes were detected in *FAE1*. *FAE1*-B03-Hap1 was mainly present in the 19 accessions with the lowest EAC (EAC $\leq$ 0.1%), whereas *FAE1*-B03-Hap2 was present in 95 accessions with high EAC (17.13% $\leq$ EAC $\leq$ 41.22%). *FAE1*-B03-Hap1 was mainly present in accessions from G4, while *FAE1*-B03-Hap2 was present mainly in accessions from G1, G2, and G3 (Figure 2c). All *FAE1*-B03-Hap1 accessions had this insertion, suggesting that this gene lost its function because of the premature termination codon produced by the insertion.

Furthermore, a two bp insertion in BjuA028960 (3-hydroxyisobutyryl-CoA hydrolase) was significantly correlated with EAC (Figure 2a,b). Comparing EAC, we found that almost all the accessions (56/57) with almost no erucic acid (EAC $\leq$ 0.01%) had this insertion, whereas it was carried only by some of the higher-EAC (1.00% $\leq$ EAC $\leq$ 42.22%) accessions (11/145). The above results suggested that this gene lost its function because of the insertion. Moreover, an eight bp insertion in BjuA028028 (methylglutaconyl-CoA hydratase, MGH) was also clearly related with EAC (Figure 2a,b). The EAC in most of the accessions with the insertion (35/58) was low (0.0001% $\leq$ EAC < 0.004%), and most of these were derived from G4 and G3. On the contrary, the EAC of all the accessions without this insertion (87) was high (11.49% < EAC $\leq$ 41.22%), mainly from G1, G2, and G3, suggesting that the premature termination codon caused by this insertion led to the loss of this gene function (Figure 2d).

In addition, through GWAS and selective elimination, we also found 192 genes significantly related to fatty acid metabolism (Table S9). These genes are conservative between G5, G6, and G7, of which 96.04% (194) were distributed in 11 regions of A08, and the rest were distributed in B03. These genes include transcription factors MYB, WRKY, GATA, and F-box/kelch-repeat proteins and are worthy of further investigation.

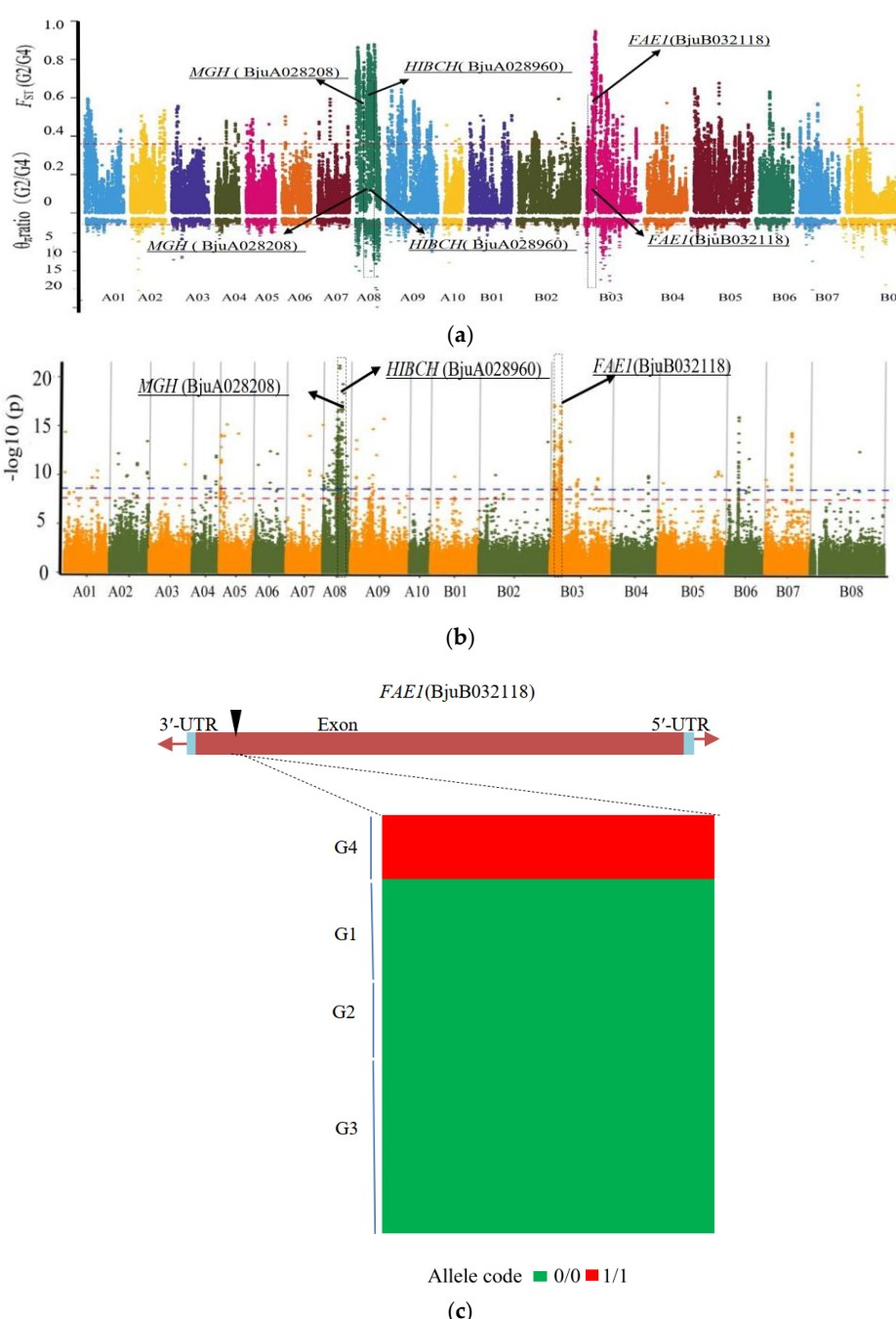

**Figure 2.** *Cont.*

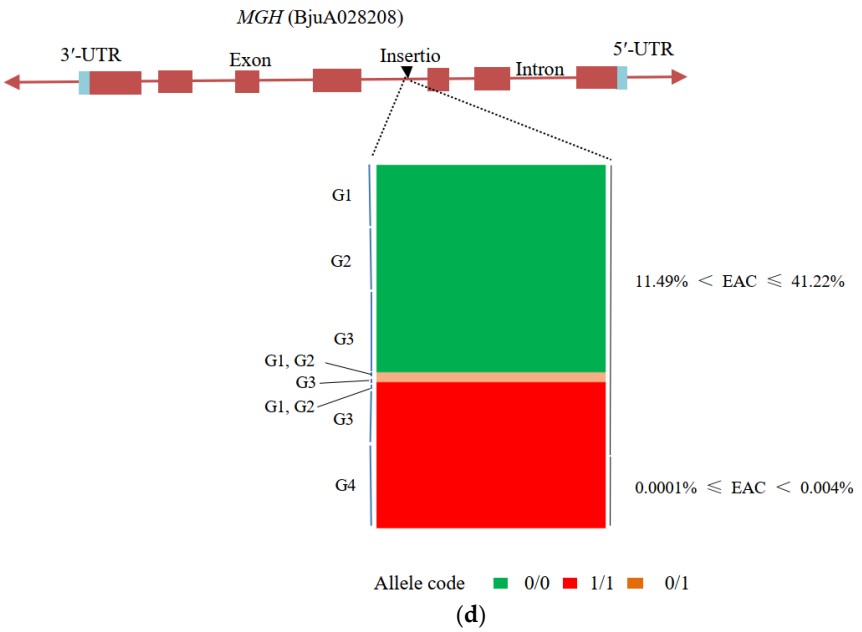

**Figure 2.** Genome-wide screening of selective sweeps and GWAS for erucic acid content in *B. juncea*. (**a**) Manhattan plot of $F_{ST}$ and $\theta_\pi$ ratio detection of selective sweeps identified from the comparisons between G3 and G5; candidate genes associated with erucic acid content are highlighted. (**b**) GWAS results of erucic acid content. Dashed horizontal lines depict the significant ($-\log_{10}(P) = 9.0$) and suggestive ($-\log_{10}(P) = 8.0$) thresholds. (**c**) Haplotypes for the candidate gene *FAE1* (BjuB032118). (**d**) Haplotypes for the candidate gene *MGH* (BjuA028028).

## 4. Discussion

In the history of crop breeding in the world, the breeding of breakthrough varieties almost always benefits from the discovery and utilization of key germplasm resources. Landraces are formed through long-term natural evolution and selection, with many excellent genetic traits. *B. juncea* has many excellent characteristics and offers a valuable gene pool for improving other *Brassica* species [11,16,46]. The Yunnan-Guizhou Plateau is the concentrated production area of *B. juncea* in China because of its diverse ecological environment (small in area, but with all types of ecosystems except desert and ocean). In this study, we collected a large genome variation data set for genetically diverse *B. juncea* accessions, which provided an opportunity to finely resolve the spread and improvement history of *B. juncea* from the Yunnan-Guizhou Plateau of China.

The studied accessions were divided into four genetically distinct groups, and extensive gene exchange occurred between the four groups. The nucleotide diversity of accessions was generally high ($\pi = 1.16 \times 10^{-3}$–$1.58 \times 10^{-3}$). These results are consistent with historical records; compared with *B. juncea* from northwest China and South Asia, *B. juncea* from southwest China shows higher nucleotide diversity ($\pi = 1.40 \times 10^{-3}$) [47], and the genetic diversity of *B. juncea* in western China is higher than that of eastern China [48]. In this study, more than 80% of the accessions came from Yunnan Province, China, but more than 98% of them had a genetic relationship coefficient of less than 0.2; in fact, 60% of them had a genetic relationship coefficient of 0. Except between G1 and G2, the degree of population differentiation between all other subgroups reached above medium level. It was supposed that this district has many mountains, complex landforms, and diverse climates, which limited the exchange and promoted the differentiation of genetic germplasms and therefore resulted in the high diversity. The utilization of these accessions with high genetic diversity will be beneficial to the study of population improvement and development history as well as the mining of related genes.

Based on the aforementioned analyses, we posit that *B. juncea* in the Yunnan-Guizhou Plateau underwent two obvious selections in its improvement process, the first of which

was mainly aimed at the yellow seed coat trait and the second mainly at low content of EAC. During the stages 2 (choosing low EAC), more genetic diversity was lost than stage1. These results are consistent with the historical breeding records in China in the last century. Many factors lead to the formation of population structure, such as geographical factors, domestication, or selection [49]. Selection is the main factor affecting the genetic structure of the plant population. If we can find out the sites in the genome related to the corresponding selection, that would be of great significance to the evaluation of germplasm resources.

Chromosome distribution analysis showed that 211 putative selective sweeps related to seed coat color were unevenly distributed on the A and B subgenomes of *B. juncea*, about 60% of which were on A09. Several researchers have shown that QTLs for seed coat color were distributed on the A09 chromosome. For example, Kebede et al. identified a major QTL and a minor QTL on the A9 chromosome of Chinese cabbage, respectively [50]. Li et al. also identified the genes related to the seed coat color of *Brassica campestris* located on the A09 chromosome and further proved that *BrTT8* gene was involved in the regulation of seed coat color [20]. Wang et al. mapped the yellow-seeded gene to the 1.04 Mb interval on chromosome A09 of *B. rapa* [51]. Liu et al. showed that a major QTL related to seed coat color of *B. napus* was also located on the A09 chromosome [52]. In addition, Padmaja et al. observed that the mutation of the *TT8* allele on the B08 chromosome led to the yellow-seeded characteristic in *B. juncea* [21]. At present, most studies focused on the A subgenomes, but few focused on the B subgenomes of *B. juncea*. The further exploration of B subgenomes will be crucial to further improve the seed coat color trait.

As a powerful way to identify genomic regions related to the agronomic traits, the GWAS has been successfully applied to understand the genetic mechanism of important agronomic traits [53,54]. Using the GWAS, Kang et al. found genes related to seed size, root expansion, and stem expansion in *B. juncea* [47]. It is still difficult to fine-map and clone the target genes that lead to the phenotypic variation of these traits. To overcome these challenges, selective elimination has recently been combined with the GWAS to identify the important genes and has acquired good results. For example, Fang et al. identified the selected regions of upland cotton in the process of domestication and located the genes related to several important agronomic traits, such as lint yield and cotton fiber quality [55]. Kang et al. found 12 candidate genes controlling the flowering stage and revealed two candidate genes, *VIN3* and *SRR1*, underlying the adaptability of *B. juncea* to the environment at the flowering stage [47]. In this work, we retrieved 3470 genes related seed coat color based on selective elimination. Of these genes, 542 were conservative in G2 and G3 subgroups. Subsequently, we combined the GWAS and selective elimination results to mine candidate genes and identified two reliable candidate genes. To maximize the detection power, researchers usually prefer the permissive model, GLM, which probably introduces more false positives than the model EMMAX. To reduce the risk of errors, this study was based on EMMAX, and these signals were verified in three different environments, implying that these correlation signals are reliable.

Proanthocyanidin (PC) is the key to determine the seed coat color, which has been verified in many crops [17,56,57]. The PC forms brown and black substances after oxidation, while the yellow-seeded accessions cannot synthesize and accumulate pigment to make the seed coat transparent and present the color of the embryo. The main pigment component of the transparent seed coat is flavonols [58–60]. After the synthesis of dihydroflavonol (DHQ), it is divided into two ways: one is oxidized by flavonol synthase (FLS) to produce flavonol substances, and the other is catalyzed by DFR to synthesize PC [61]. FLS and DFR compete for the common substrate, DHQ. In the second pathway, 4-dihydroflavonol reductase (DFR), anthocyanin synthase (ANS), and anthocyanin reductase (ANR, or BAN) are considered to be the key enzymes in the PC synthesis pathway [62,63], and these enzymes are positively regulated by transcription factors (TFs) such as *TT2* (MYB TF), *TT8* (bHLH TF), and *TTG1* (WD40 TF) [64,65].

The abnormal expression of DFR, ANS, ANR, and related positive regulatory TFs will lead to the transparent seed coat (yellow-seeded trait). However, this study failed

to confirm the above results, which may be due to the difference in genetic background of the materials; most of the previous research materials are DH populations or hybrid populations produced by the cross between black-seeded and yellow-seeded masteries, and a natural population was adopted for the GWAS in this study. In combination with the GWAS and selective elimination, two reliable candidate genes (BjuB018407 and BjuA034936) related to seed coat color were identified in this study. BjuB018407encodes FLS, the key enzyme of flavonol synthesis in the second pathway.

The color of plant flowers is partly determined by the competition between FLS and DFR for the common substrates [66,67]. Overexpression of the *DFR* gene in tobacco promotes anthocyanin synthesis [68], and silencing of the *FLS* gene in tobacco results in significantly upregulated expression of *DFR* and *ANS* [69]. When *McFLS* is overexpressed or *McDFR* is silenced, the flavonol content in begonia leaves and apple peels increases [66]. Although there is a competitive relationship between FLS and DFR, some studies show that FLS has priority in the competition for their substrate, DHQ [70]. The above studies show that the competitive relationship between FLS and DFR affects the accumulation of PC and flavonoid alcohols and then affects the seed coat color. In this study, two missense mutations detected in the BjuB018407 gene encoding FLS were significantly related to the seed coat color, and more than 96% of the yellow-seeded materials carried these mutations. It is speculated that these mutations may affect the competitive power of FLS on the substrate DHQ and then lead to the production of the transparent seed coat.

In the plant's fatty acid biosynthesis pathway, *FAE1* is the rate-limiting enzyme of the carbon chain elongation reaction and and a key enzyme for the biosynthesis of ultra-long chain fatty acids more than 20 carbon atoms [71]. The mutation of this gene leads to the blocking of erucic acid synthesis [32,33], resulting in low-erucic-acid rapeseed varieties. The world's first low-erucic-acid rapeseed variety, Oro, was cultivated based on a point mutation at the 845th base (282nd position of the amino acid sequence) of *FAE1*; the corresponding amino acid at this position changed from serine to phenylalanine, and the EAC in rapeseed seeds decreased sharply [27,72]. This variety has set a precedent for low-erucic-acid breeding in rapeseed, and many low-erucic-acid rapeseed varieties have been selected in succession. However, a large number of studies have shown that the traits of low EAC in most rapeseed varieties are also caused by the single amino acid variation at the 282nd position of *FAE1*, which is the same as Oro [27,73–76]. Meanwhile, some scholars have reported a new low EAC mutation independent of the above point mutation, which is caused by the deletion of a four-base pair of the *FAE1* gene [77,78]. In short, the low EAC mutation types in rapeseed are relatively few. A new type of *FAE1* mutation was found in this study. The insertion of 1 bp at the 9th base led to a code shift variant and obtained a stop code. In this study, the EAC of materials carrying this mutation was the lowest and less than 0.1%. In addition, this study also found that the frameshift mutation of the genes encoding 3-hydroxyisobutyryl-CoA hydrolase and methylglutaryl coenzyme A hydrolase were also significantly related to low EAC, but there were relatively few relevant reports, which need further verification.

## 5. Conclusions

Our study provides a theoretical basis for the division of germplasm protection units as wells as the evaluation and utilization of excellent germplasm from *B. juncea* in southwest China. This study indicated that *B. juncea* from southwest China shows higher nucleotide diversity and provides valuable data for understanding the improvement history of *B. juncea* at the Yunnan-Guizhou Plateau. *B. juncea* from southwest China has undergone two obvious selections with the loss of genetic diversity, especially in the improvement process of selecting low erucic acid. A selective sweep analysis and a genome-wide association study revealed the candidate genes for seed color and fatty acid biosynthesis. In addition, a new type of mutation, the insertion of 1 bp at the ninth base of *FAE1*, led to the trait of low EAC. The significant SNPs associated with favorable variants and candidate genes will be a valuable resource for further improving the seed coat color and seed quality.

**Supplementary Materials:** The supporting information can be downloaded at: https://www.mdpi.com/article/10.3390/agronomy13041053/s1, Figure S1: Single nucleotide polymorphisms (SNP) and InDels distribution in the 18; Figure S2: Phylogenetic tree of 193 *B. juncea* accessions; Figure S3: All accessions and group-specific LD decay plots; Figure S4: The heatmap of kinship of 193 *B. juncea* accessions; Figure S5: Seed coat color of 193 *B. juncea* accessions; Figure S6: Frequency distribution diagram of erucic acid content; Figure S7: GWAS of erucic acid metabolism in *B. juncea* in three years. Candidate genes significantly associated with erucic acid metabolism are labeled in each year. The significance threshold of $-\log10 p$ value was set at 8 in 193 *B. juncea* accessions chromosomes of *B. juncea*; Table S1: Information for the 193 *Brassica juncea* accessions used in this study; Table S2: Chromosomal distribution of SNPs and Indels in the 193 *Brassica juncea* accessions; Table S3: Statistics of fixation index (FST) and nucleotide diversity ($\pi$); Table S4: Seed coat color candiate genes within the putative selective sweeps in *Brassica juncea*; Table S5: Genes significantly associated with seed coat color in *Brassica juncea*; Table S6: Phenotypic characteristics for fatty acid compositiont in the 193 *Brassica juncea* accessions; Table S7: Erucic acid metabolism candiate genes within the putative selective sweeps in *B. juncea*; Table S8: Genes significantly associated with erucic acid metabolism in *Brassica juncea*; Table S9: List of 192 genes significantly related to erucic acid metabolism.

**Author Contributions:** Formal analysis, X.Y., C.Q. and F.L.; investigation, X.Y., M.F., G.L., C.Q., H.L., X.L., Y.Z. (Yunyun Zhang), Y.Z. (Yusong Zhang), K.Z., L.Z., Y.L., J.L., X.H., L.H. and F.L.; project administration, X.Y. and F.L.; resources, G.L., Y.Z. (Yunyun Zhang) and K.Z.; supervision, X.Y. and F.L.; validation, X.Y., M.F. and X.L.; writing—original draft, X.Y., M.F., C.Q., H.L., X.L., Y.Z. (Yunyun Zhang) and F.L.; writing—review and editing, X.Y. and F.L. All authors have read and agreed to the published version of the manuscript.

**Funding:** This work was supported by grants from the National Natural Science Foundation of China (31960410), major science and technology projects of Yunnan Province (202102AE090002), and the National Rape Industry Technology System of China (CARS-12).

**Data Availability Statement:** Data generated during the study are within the article or Supplementary Materials.

**Acknowledgments:** We acknowledge excellent technical assistance from Jingqiao Wang.

**Conflicts of Interest:** The authors declare no conflict of interest.

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
