# Peer review of "Whole-Genome Resequencing Reveals the Genetic Diversity and Selection Signatures of the Brassica juncea from the Yunnan-Guizhou Plateau"

_agronomy, doi:10.3390/agronomy13041053_

Round 1
Reviewer 1 Report
In line 102, there should be a space between “2021” and “growing”.
In line 163, please explain why the threshold 1e-8 was chosen. Do you think the threshold using the Bonferroni Correction is better? Are the results more reliable if more GWAS methods are added?
In line 160, the introduction of method and software for calculating kinship matrix should be added.
In line 343, the authors mentioned selective elimination. Please explain how to implement it.
In line 166, is it better to use a region within a specific distance of each significant SNP to search genes?
Author Response
Response to Reviewer 1 Comments
Dear Editors and Reviewers:
Thank you for your letter and for the reviewers’ comments concerning our manuscript entitled “Whole-genome resequencing reveals the genetic diversity and selection signatures of the Brassica juncea from Yunnan-Guizhou Plateau” (agronomy-2268450). Those comments are all valuable and very helpful for revising and improving our paper, as well as the important guiding significance to our research. We have studied comments carefully and have made correction which we hope meet with approval. Revised portions are marked in red in the paper. The main corrections in the paper and the responds to the reviewer’s comments are as flowing:
Responds to the reviewer’s comments:
Point 1: In line 102, there should be a space between “2021” and “growing”.
Response 1: We 've corrected the error. We apologize for our error.
Point 2: In line 163, please explain why the threshold 1e-8 was chosen. Do you think the threshold using the Bonferroni Correction is better? Are the results more reliable if more GWAS methods are added?
Response 2: The analysis of GWAS and other genome-wide statistical tests require a method to correct the threshold of results. Bonferroni correction is a rigorous multiple test correction method. Bonferroni correction divides the set significance level by the number of tests to obtain a total threshold value. In this study, log10 (0.05/3972292) =8.0. The application of Bonferroni correction usually provides us with the most conservative p-value threshold, but often leads to false negatives. In practice, the threshold is adjusted based on the Manhattan map. In this article, we conducted a comprehensive assessment of 1e-8-le10.
We agree with you that the more GWAS analysis methods, the more reliable the results will be. During the actual analysis process of this study, GLM, EMMAX, and FASTLMM methods were used to conduct GWAS analysis on some traits. The results showed that the false positive rate of GLM was high, and the results of FASTLMM and EMMAX were highly consistent. In addition, we compared the results of three growth cycles for all traits and found that the repeatability was very good, indicating that the results were more reliable. If the condition allows, more GWAS analysis methods will be used for comparison and verification, in the future.
Point 3: In line 160, the introduction of method and software for calculating kinship matrix should be added.
Response 3: We have added the method and software for calculating kinship matrix in “2.3. Population structure and phylogenetic analyses”, “The kinship matrix (K) of the association population was calculated using TASSEL 5.2.1.”
Point 4: In line 343, the authors mentioned selective elimination. Please explain how to implement it.
Response 4: When favorable mutations occur, the higher the fitness of the mutant gene, the easier it is to be selected and fixed. The chromosomal regions connected to this locus are also fixed, and a large number of closely connected chromosomal regions lose polymorphism. Selective elimination regions are regions where genomic diversity is reduced and high frequency alleles exist. In this study, there were significant differences in seed coat color and erucic acid content among the several subgroups of test materials. Taking erucic acid metabolism as an example, in order to detect candidate regions such as fatty acid metabolism that may be affected by selection, nucleotide diversity (π) and fixation index (FST) between paired subgroups were calculated. According to FST and θπ ratio, select selective scanning regions related to fatty acid metabolism between different subgroups. In the comprehensive evaluation of FST and θπ ratio, a 5% intersection of the two indicators is selected as the signal selection. Annotated genes located in these regions are considered candidate genes, and further combined with GWAS analysis results to screen candidate genes.
Point 5: In line 166, is it better to use a region within a specific distance of each significant SNP to search genes?
Response 5: Agree with your opinion, we searched genes within a specific region of the SNP. Previously, the expression in the method section was inaccurate, and this section has been revised.” The significant thresholds of all tested traits were evaluated with the formula P= 0.05/n (where n is the number of SNPs). Then, 100 kb sequence regions adjacent to the significantly associated SNPs were searched for associated genes.”
We appreciate for your warm work earnestly and hope that the correction will meet with approval.
We would like to thank you again for taking the time to review our manuscript.

Reviewer 2 Report
The present manuscript describes the analysis of genetic diversity of Brassica juncea by GWAS based on Whole-Genome Resequencing. The authors also used the resulted SNPs to identify candidate genes for seed colour and fatty acid biosynthesis. While genotyping was well performed and described, there are some problems (described below) with phenotypic data that should be improved or clarified. This is important since the seed color and erucic acid traits are the basis of the results and discussion.
In order to accept this paper for publication a major revision is needed.
Introduction:
English should be improved, there are mistakes such as:
"-The records of its center origin are diversity"- should be diverse.
-As we known - as we know….
-2021growing season: should be 2021 growing seasons.
-“The Yunnan-Guizhou plateaus are one of the districts possessing highest genetic diversity of cultivated B. juncea in China. Should add: In this study we sequenced 193 B. juncea accessions…add most from the Yunnan-Guizhou plateaus
Traits:
1. Table 1. and Table S5: the author should show normal distribution of each trait in the whole collection. Traits which are not normally distributed are problematic for GWAS.
2. There is very high variation among lines in some traits (CV% around 60).
3. There are values of 0.00 (e.g. arachidonic acid and erucic acid), were they exclude from the GWAS analysis (NA?)?
4. How authors can explain that the same maximum value (11.53) of Palmitic acid were obtained in the 3 years? there are some values that have higher SD than mean(4.07±9.92 , 3.50±8.49).
5. Table S1 – seed color- “…. G4 contained mostly of the black-seeded accessions, G6 and G7 almost included only the yellow-seeded accessions, while G2, G3 and G5 consisted mostly of the yellow-seeded ones”. Table 1 includes only brown and yellow seeds, it does not include black seeded accessions. The division between yellow and brown should be accurate. If the authors wish to show that clusters differ between traits, they should provide values of statistical tests showing significant differences between clusters for this specific trait and also for other traits.
6. Was this information for seed color “2 present yellow and 1 present brown” was used for phenotyping of 193 accessions in the GWAS analysis?
Statistical analysis
“Statistical analyses of phenotypic data were performed with the SPSS soft”. Please add information on the type/name of statistical test (within the SPSS).
“The PCA includes relatively scattered spatial distribution characteristics. Also, the PCA plot show very low values. For interpretation of PCA I recommend reading 1471-2164-15-767 (1).pdf
STRUCTURE – please add detail, how many burning periods were made for K-clusters
What are the “two materials”: ”The kinship value between each two materials were calculated”.
Line 193: “The results indicate that there exists genetic crossover phenomenon among all the subgroups”. Please explain.
Author Response
Response to Reviewer Comments
Dear Editors and Reviewers:
Thank you for your letter and for the reviewers’ comments concerning our manuscript entitled “Whole-genome resequencing reveals the genetic diversity and selection signatures of the Brassica juncea from Yunnan-Guizhou Plateau” (agronomy-2268450). Those comments are all valuable and very helpful for revising and improving our paper, as well as the important guiding significance to our research. We have studied comments carefully and have made correction which we hope meet with approval. Revised portions are marked in red in the paper. The main corrections in the paper and the responds to the reviewer’s comments are as flowing:
Responds to the reviewer’s comments:
Point 1: The records of its center origin are diversity"- should be diverse.
Response 1: We’ve changed “The records of its center origin are diversity. “to “The records of its center origin are diverse.”
Point 2: As we known.
Response 2: We’ve changed “As we known” to “As we know”.
Point 3: 2021growing season: should be 2021 growing seasons.
Response 3: We’ve changed “growing season” to “growing seasons”.
Point 4: The Yunnan-Guizhou plateaus are one of the districts possessing highest genetic diversity of cultivated B. juncea in China. Should add: In this study we sequenced 193 B. juncea accessions…add most from the Yunnan-Guizhou plateaus
Response 4: We’ve changed “Herein, in this study we sequenced 193 B. juncea accessions to examine their intraspecific genetic diversification and detected a series of candidate genes for seed coat color and erucic acid by using the selective sweeps and genome-wide association studies (GWAS).” to “Herein, in this study we sequenced 193 B. juncea accessions ,most of which from the Yunnan-Guizhou plateau to examine their intraspecific genetic diversification and detected a series of candidate genes for seed coat color and erucic acid by using the selective sweeps and genome-wide association studies (GWAS).”.
Point 5: Table 1. and Table S5: the author should show normal distribution of each trait in the whole collection. Traits which are not normally distributed are problematic for GWAS.
Response 5: According to your suggestion, we have made modifications. The kurtosis and skewness values of all traits have been added in table 1. The frequency distribution diagram of erucic acid content is added in Supplementary Figure 6. The erucic acid content of 193 B. juncea accessions generally conformed to a normal distribution.
Point 6: There is very high variation among lines in some traits (CV% around 60).
Response 6: The coefficient of variation reflects the degree of dispersion of data. Due to the large genetic differences of 193 accessions in this study, especially the contents of erucic acid, the coefficient of variation of the overall material is relatively high. Some low erucic acid materials have an erucic acid content of less than 0.1%, and some have an erucic acid content exceeding 40%. After subgroup classification, the coefficients of variation within each subgroup are mostly between 10% and 30%.
Point 7: There are values of 0.00 (e.g., arachidonic acid and erucic acid), were they exclude from the GWAS analysis (NA?)?
Response 7: These values were not excluded from GWAS. The reason why these values were displayed as zero was mainly because all previous numbers retain only two decimal places and were rounded to zero, which was actually greater than zero. The number of decimal places had been increased to four decimal places in the corresponding table.
Point 8: How authors can explain that the same maximum value (11.53) of Palmitic acid were obtained in the 3 years? there are some values that have higher SD than mean (4.07±9.92 , 3.50±8.49).
Response 8: The reason why the same value occurs is also because the values only retain two decimal places after the decimal point, and the values after rounding are the same. Changes have been made to retain four decimal places.
Some values have high SD values, on the one hand due to the large genetic differences in the materials selected in this study, and on the other hand due to the unreasonable classification of previous subgroups. Through reanalysis of the data, the subgroups were reclassified, and all data were submitted for reanalysis. Currently, the SD values for all traits are more reasonable than before (Table1).
Point 9: Table S1 – seed color- “…. G4 contained mostly of the black-seeded accessions, G6 and G7 almost included only the yellow-seeded accessions, while G2, G3 and G5 consisted mostly of the yellow-seeded ones”. Table 1 includes only brown and yellow seeds; it does not include black seeded accessions. The division between yellow and brown should be accurate. If the authors wish to show that clusters differ between traits, they should provide values of statistical tests showing significant differences between clusters for this specific trait and also for other traits.
Response 9: We are very sorry. Here is a writing error. We’ve changed “G4 contained mostly of the black-seeded accessions” to “G4 contained mostly of the brown-seeded accessions”. We also agree with reviewer’s suggestions that the division between yellow and brown should be accurate. In this study, the color difference between yellow-seeded and brown-seeded were significant. Previously, in supplementary figure 3, only the grain photos of each material were listed, without subgroup classification, and the color difference was not significant. After modification, we provided photos of seeds classified by subgroup.
Point 10: Was this information for seed color “2 present yellow and 1 present brown” was used for phenotyping of 193 accessions in the GWAS analysis?
Response 10: Considering the color difference between yellow and brown grains in this study is large, the GWAS analysis of the seeds color is considered as a classification trait, with values of 0 and 1 assigned to yellow and brown. Previously, using 1 and 2 to represent grain color was indeed inappropriate and prone to ambiguity. We have made changes to make it clearer to use language descriptions combined with pictures (supplementary Figure 5).
Point 11: “Statistical analyses of phenotypic data were performed with the SPSS soft”. Please add information on the type/name of statistical test (within the SPSS).
Response 11: We have added the information on the type/name of statistical test in line 109 as required,” Descriptive statistics, analysis of variance (ANOVA) and correlation analysis of phenotypic data were performed using IBM SPSS Version 24.”
Point 12: “The PCA includes relatively scattered spatial distribution characteristics. Also, the PCA plot show very low values. For interpretation of PCA I recommend reading 1471-2164-15-767 (1).pdf
Response 12: We have revised the interpretation of PCA with reference you recommended。
Point 13: STRUCTURE – please add detail, how many burning periods were made for K-clusters.
Response 13: In this study, we used ADMIXTURE software to analyze population structure. ADMIXTURE uses the same model as STRUCTURE and runs more efficiently.
Point 14: What are the “two materials”:” The kinship value between each two materials were calculated”.
Response 14: We’ve changed “The kinship value between each two materials were calculated” to “The kinship value between each two of 193 accessions were calculated”.
Point 15: Line 193: “The results indicate that there exists genetic crossover phenomenon among all the subgroups”. Please explain.
Response 15: The word "genetic crossover" is incorrect. We’ve changed “genetic crossover” to “genetic exchange”.
We appreciate for your warm work earnestly and hope that the correction will meet with approval.
In addition, the language has also been revised again by the author, who is better at both Chinese and English.
We would like to thank you again for taking the time to review our manuscript.

Reviewer 3 Report
I have reviewed the article entitled 'Whole-Genome Resequencing Reveals the Genetic Diversity and Selection Signatures of the Brassica juncea from Yunnan-Guizhou Plateau', which has been submitted to the journal Agronomy.
In this research article, the genomes of 193 indigenous B. juncea accessions were re-sequenced, obtaining 1.04 million high-quality SNPs and 3.23 million InDels by mapping reads to the reference genomes of B. juncea var. timuda. Population genetic structure and phylogenetic analyses were performed. Additionally, selective sweep analysis and a genome-wide association study were carried out.
The article is interesting from a genetic standpoint and may provide insights into the spread and improvement of B. juncea. However, I have a few concerns after this first round of review.
Population structure: Seven populations were inferred, but other methods, such as the Evanno test and/or Deviance Information Criterion (DIC), should be used to confirm these results. A dendrogram (based on UPGMA, for instance) with bootstrapping would also be desirable. Additionally, figure 1 and Suppl. Figure with NJ are not convincing (and it has not bootstrap values, and it was not reported). Please consider redoing the analysis using different methods and include this information in the main text, not suppl.
The results in Table 1 are unnecessary if they do not take statistics into account. A deeper statistical analysis is needed, including multiple comparison tests.
A plot of LD decay with the complete genome is necessary (not in separate groups).
For the GWAS, why was the significance threshold of -log10 p value set at 8? What kind of correction was done? False discovery rate should be used.
Author Response
Response to Reviewer Comments
Thank you for your letter and for the reviewers’ comments concerning our manuscript entitled “Whole-genome resequencing reveals the genetic diversity and selection signatures of the Brassica juncea from Yunnan-Guizhou Plateau” (agronomy-2268450). Those comments are all valuable and very helpful for revising and improving our paper, as well as the important guiding significance to our research. We have studied comments carefully and have made correction which we hope meet with approval. Revised portions are marked in red in the paper. The main corrections in the paper and the responds to the reviewer’s comments are as flowing:
Responds to the reviewer’s comments:
Point 1: Population structure: Seven populations were inferred, but other methods, such as the Evanno test and/or Deviance Information Criterion (DIC), should be used to confirm these results. A dendrogram (based on UPGMA, for instance) with bootstrapping would also be desirable.
Response 1: According to your suggestion, we have conducted a new subgroup classification based on population structure analysis, phylogenetic, principal component analysis, and material phenotypic traits. Several analysis results consistently indicate that the optimal subgroup is 4 (Figure 1b, 1c, 1d).
Point 2: Additionally, figure 1 and Suppl. Figure with NJ are not convincing (and it has not bootstrap values, and it was not reported). Please consider redoing the analysis using different methods and include this information in the main text, not suppl.
Response 2: Based on your comments, we have re conducted a phylogenetic analysis and placed the analysis results in the text (Figure 1d). Clades with bootstrap value of above 50% are indicated by a circled red dot.
Point 3: A plot of LD decay with the complete genome is necessary (not in separate groups).
Response 3: We have modified and added a plot of LD decay with the complete genome (Supplementary Figure 3).
Point 4: For the GWAS, why was the significance threshold of -log10 p value set at 8? What kind of correction was done? False discovery rate should be used.
Response 4: The analysis of GWAS and other genome-wide statistical tests require a method to correct the threshold of results. Bonferroni correction is a rigorous multiple test correction method. Bonferroni correction divides the set significance level by the number of tests to obtain a total threshold value. In this study, log10 (0.05/3972292) =8.0. The application of Bonferroni correction usually provides us with the most conservative p-value threshold, but often leads to false negatives. In practice, the threshold is adjusted based on the Manhattan map. In this article, we conducted a comprehensive assessment of 1e-8-le10.
Point 5: The results in Table 1 are unnecessary if they do not take statistics into account. A deeper statistical analysis is needed, including multiple comparison tests.
Response 5: Based on your valuable comments, we have conducted deeper statistical analysis of the phenotypic data, adding some statistical analysis, including multiple comparisons (Table1).
We appreciate for your warm work earnestly and hope that the correction will meet with approval.
We would like to thank you again for taking the time to review our manuscript.

Round 2
Reviewer 3 Report
The authors have addressed most of my comments. The 4 subpopulations found are much more reasonable, taking into consideration that the amount of cluster in a crop obeys a Poisson distribution. With k=4 , all results have changed, and therefore interpretation may change. Authors should be careful with this.
I suggest that in several parts of the text, change cluster (or subgroup) to "genetically distinct groups".
I suggest authors that they rethink the conclusions, since they did not change when a new analysis was made. This is not in tune with science, since no matter the results, will it always be the same conclusion? This is indicating that the conclusions are extremely general, and that is useless, since it can be written for any article or experiment. That is, they must say something about the general characteristics of the candidate genes, or in which process they participate mostly (for this, see the results of R2, or the coefficients of variation explained by the markers. Polish the results more, and conclude accordingly.
Author Response
Response to Reviewer Comments
Thank you again for your suggestions for our manuscript entitled “Whole-genome resequencing reveals the genetic diversity and selection signatures of the Brassica juncea from Yunnan-Guizhou Plateau” (agronomy-2268450). Your comments are very helpful in improving my article. We have studied comments carefully and have made correction which we hope meet with approval. Revised portions are marked in red in the paper. The main corrections in the paper and the responds to the reviewer’s comments are as flowing:
Responds to the reviewer’s comments:
Point 1: I suggest that in several parts of the text, change cluster (or subgroup) to "genetically distinct groups".
Response 1: According to your suggestion, we have change "subgroup to "genetically distinct groups” in thirteen parts of the text.
Point 2: I suggest authors that they rethink the conclusions, since they did not change when a new analysis was made. This is not in tune with science, since no matter the results, will it always be the same conclusion? This is indicating that the conclusions are extremely general, and that is useless, since it can be written for any article or experiment. That is, they must say something about the general characteristics of the candidate genes, or in which process they participate mostly (for this, see the results of R2, or the coefficients of variation explained by the markers. Polish the results more and conclude accordingly.
Response 2: Based on your valuable comments, we have carefully rethought the conclusions. The modified conclusion is as follows, "Our study provides a theoretical basis for the division of germplasm protection units, the evaluation and utilization of excellent germplasm from B. juncea in southwest China." This study indicated that B. juncea from southwest China shows higher nucleotide diversity and provides valuable data for understanding the improvement history of B. juncea at Yunnan-Guizhou Plateau. B. juncea from southwest China had undergone two obvious selections with the loss of genetic diversity, especially in the improvement process of selecting low erucic acid. Selective sweep analysis and a genome-wide association study revealed that candidate genes for seed color and fatty acid biosynthesis. In addition, a new type of mutation the insertion of 1 bp at the 9th base of FAE1 led to the traits of low EAC. The significant SNPs associated with favorable variants and candidate genes will be a valuable resource for further improving the seed coat color and seed quality.”
We appreciate for your warm work earnestly and hope that the correction will meet with approval.
We would like to thank you again for taking the time to review our manuscript.
